# Challenges to Mitigating the Urban Health Burden of Mosquito-Borne Diseases in the Face of Climate Change

**DOI:** 10.3390/ijerph18095035

**Published:** 2021-05-10

**Authors:** Antonio Ligsay, Olivier Telle, Richard Paul

**Affiliations:** 1The Graduate School, University of Santo Tomas, Manila 1008, Philippines; adligsay@yahoo.com; 2Clinical and Health-Related Research, St. Luke’s Medical Center WHQM College of Medicine, Quezon City 1112, Philippines; 3CNRS, Géographie-Cités, Paris 1 Université Paris-Sorbonne, 75006 Paris, France; telle.olivier@gmail.com; 4Functional Genetics of Infectious Diseases Unit, Institut Pasteur, UMR 2000 (CNRS), 75015 Paris, France

**Keywords:** climate change, urban heat islands, mosquito-borne disease, mitigation strategies

## Abstract

Cities worldwide are facing ever-increasing pressure to develop mitigation strategies for all sectors to deal with the impacts of climate change. Cities are expected to house 70% of the world’s population by 2050, and developing related resilient health systems is a significant challenge. Because of their physical nature, cities’ surface temperatures are often substantially higher than that of the surrounding rural areas, generating the so-called Urban Heat Island (UHI) effect. Whilst considerable emphasis has been placed on strategies to mitigate against the UHI-associated negative health effects of heat and pollution in cities, mosquito-borne diseases have largely been ignored. However, the World Health Organization estimates that one of the main consequences of global warming will be an increased burden of mosquito-borne diseases, many of which have an urban facet to their epidemiology and thus the global population exposed to these pathogens will steadily increase. Current health mitigation strategies for heat and pollution, for example, may, however, be detrimental for mosquito-borne diseases. Implementation of multi-sectoral strategies that can benefit many sectors (such as water, labor, and health) do exist or can be envisaged and would enable optimal use of the meagre resources available. Discussion among multi-sectoral stakeholders should be actively encouraged.

## 1. Introduction

Five years on from the landmark Paris Agreement in 2015 to reduce global warming to below 2 °C, global carbon dioxide emissions have continued to rise steadily, with a resulting increase in the global average temperature of 1.2 °C above pre-industrial levels [1,2]. With the five hottest years ever recorded coming after 2015, global warming continues to accelerate. Impacts of climate change are omnipresent and can be observed across the different sectors (agriculture, energy, water supply, coasts, ecosystems, forests, society, transport, and health). They are, however, unevenly distributed across the globe, and, consequently, there are large regional differences in the exposure, vulnerability, and adaptation to climate change, and impacts are disproportionally occurring in countries not responsible for our plight [2]. Climate change is already impacting human health with the burden expected to increase over the coming decades [3]. Climatic changes have devastating impacts on many aspects of health through droughts, floods, storms, coastal flooding, forest fires, agricultural production, natural water sources, landslides, heat waves, and the proliferation of microbes and spread of arthropod vectors of pathogens [4,5,6,7,8]. Analyses suggest that low-income countries will bear the brunt of the predicted health impact [2,9,10]. Africa and Asia, for example, bear the largest economic burden of disease in humans [11] and are where the influence of climate variability on health is already widely recognized [12,13,14]. Managing health risks will require modifying health systems to become more resilient. Such adaptation will be required for decades, with the degree of mitigation being a key determinant of the ability of the health systems to manage risks projected later into the century [3,15]. Perseverance with a business-as-usual approach to climate change will endanger lives and livelihoods, leading to a higher health burden that could have been prevented. While the time scale of climate change is in the order of decades, decision frameworks for public health officials and regional planners need to be based on much shorter time scales [16]. The ensuing costs of managing the health risks of climate variability will be very significant including not only costs associated with increased health care and public health interventions, but also costs associated with the labor sector (lost work days and lower productivity) and with maintaining well-being; this latter will be particularly pronounced in the urban setting. Cities are particularly vulnerable to climate-associated health hazards and deserve particular attention with respect to mitigation strategies for both non-infectious and infectious diseases.

The aim of this paper is to provide an overview to describe the health issues likely to be exacerbated by climate change specifically in an urban context and both why and how urban mosquito-borne diseases need to be put at the center of the agenda of implementing mitigation strategies for urban health.

## 2. Urbanization and Health

The most important anthropogenic influences on the climate are greenhouse gas emissions and land use change, notably urbanization where vegetation is replaced by man-made surfaces [17,18,19]. The growth of urban populations and the large-scale migration of individuals among locations are among the key defining environmental challenges of the 21st century, especially for human health. Over 50% of the world’s population live in cities, and this is expected to increase to approximately 70% by 2050 [20]. Cities occupy only 3% of land surface [21] and yet they produce approximately 80% of the gross world product, consume about 78% of the world’s energy, and produce more than 60% of all CO_2_ emissions [22,23,24]. The expected dramatic increase in urbanization precludes extrapolation from our current limited understanding of urban health to the future, especially at relevant resolutions taking into account pronounced demographic, socio-economic, environmental, and climatic heterogeneity. Socially disadvantaged individuals living in urban settings have been highlighted as a major group at risk for the adverse health consequences of climate change, which will exacerbate the currently recognized urban health disparities [25,26,27].

The extensive production of greenhouse gases and urbanization both lead to increases in surface temperatures and can lead to urban heat islands (UHIs), metropolitan areas that are significantly warmer than the surrounding rural areas [28]. Many factors contribute to the creation of UHIs: population density, percent built-up area and density, the reflectivity (albedo), thermal bulk properties of man-made surfaces, including impervious surfaces (roads, pavements), lack of vegetation areas and water bodies, the thermal mass that is produced by anthropogenic activities (transportation, industry), and urban morphology (high-rise building, variation in the height of building, sky view factor, etc.) [29,30]. Whilst generally considered at the city vs. neighboring rural environment scale, the effects of urban geometry, both with the shading effect in daytime and with the reduction of radiative cooling and increasing thermal storage at night, can generate differences in UHI intensity at a very local scale [31,32].

The public health impact of UHIs has been directly implicated in exacerbating the negative effects of extreme temperature conditions, air pollution, poorer water quality, and general discomfort [33,34]. Increasing temperatures will inevitably lead to hotter days, more frequent and longer heatwaves leading to an increase in heat-related deaths. An increase in particulate matter is also anticipated, not least from an increase in wildfires. This will lead to an increase in Chronic Obstructive Pulmonary Disease and Cardio-vascular Disease. Finally, air quality will worsen with respect to airborne allergens, notably pollen from plants where the season will likely be extended, leading to an increase in allergic diseases, notably asthma and rhino-conjunctivitis. In addition to the evident impact on agriculture in rural areas, 90% of cities are coastal, and thus an increase in extreme weather events, such as floods and storms, will have a disproportionally high impact on the urban environment. Mental health will also decline, an effect particularly felt in urban environments, due to the cumulative effects of increased heat. In addition to the negative consequences of global warming for non-infectious diseases, infectious diseases, particularly in an urban setting, are also a cause for concern [35,36,37,38,39,40,41]. Global warming will lead to a reduction in water quality and food safety with a proliferation of microbes and an increase in bacterial and viral contamination of water and food. Last but not least, as expanded upon below, urban mosquito-borne diseases pose a particular problem and one not currently significantly addressed within the World Health Organization (WHO) Urban Health Initiative for integrating health in urban and territorial planning [42]. There is increasing documentation of increased dengue risk associated with intra-urban UHIs, generated by high population density and built-up area, disproportionately affecting the socio-economically fragile population. This is likely to be the case for all mosquito-borne diseases [30,43,44].

## 3. Urban Mosquito-Borne Diseases

Many of the most important diseases affecting health are mosquito-borne, notably malaria and the arboviral diseases (caused by viruses of the Flaviviridae, Bunyaviridae and Togaviridae viral families). The WHO estimates that one of the main consequences of global warming will be an increased burden of such vector-borne diseases. Many of the mosquito-borne pathogens currently limited to the tropics and sub-tropics are anticipated to extend their current geographical ranges, invading more temperate regions [45]. Globalization and increased international travel and trade will enable the unwitting importation of invasive mosquito vectors and mosquito-borne pathogens from endemic areas into formerly disease-free regions, given that infections following infection by many such pathogens can be symptom-free [46,47,48]. This potential for the spread of mosquito-borne diseases into temperate regions has already occurred with dengue or the chikungunya virus transmission detected sporadically in France [49,50], Madeira [51], and Croatia [52]; and there have been frequent outbreaks in the USA [53,54], where historically both malaria and yellow fever epidemics occurred frequently until the middle of the 20th century [55,56,57].

Temperature modifications can have a large impact on the epidemic potential of mosquito-borne diseases directly through the influence of the temperature on the developmental rates of both the mosquito and the pathogen within the mosquito, as well as on mosquito survival [58]. Determining whether climate change has an actual role in the spread of pathogens remains controversial [59,60,61,62]. However, the extent to which global warming actually impacts the mosquito-borne disease burden may be of little relevance for the urban setting. The UHI effect can generate local temperatures that are far beyond any projected global temperature increases, even under the worst-case scenario. There are an increasing number of fine scale studies implicating an increased incidence of dengue, for example, and mosquito vector densities with land surface temperature, vegetation indices, and vertical cities at an intra-urban scale [29,43,44]. Thus, for a variety of reasons, urbanization will be associated with increased local temperatures compared to neighboring rural areas, with a resulting impact on the local burden of mosquito-borne diseases [30]. Climate-associated changes in precipitation will also be expected to disproportionately affect the urban and rural environments with respect to the availability of mosquito oviposition sites, with urbanization recognized to create diverse aquatic habitats for mosquito larvae [63]. Thus, global projections of mosquito species distributions and abundance and the subsequent potential exposure to associated pathogens may poorly estimate the urban condition.

Several of the mosquito-borne diseases are of particular concern, as the mosquito species in question adapt to the urban environment. This is the case for *Aedes aegypti*, the vector of dengue, Zika, chikungunya and yellow fever viruses, *Anopheles stephensi*, the vector of urban malaria (caused by either *Plasmodium falciparum* or *Plasmodium vivax*), and *Culex* spp., vectors of West Nile virus. These pathogens are inflicting a huge health burden globally. More than 3.5 billion people are at risk of dengue virus (DENV) infection in over 100 countries and recent estimates suggest that there are 390 million DENV infections every year, of which 100 million cause clinical symptoms [64]. The explosive 2015–2016 Zika epidemic infected over a million individuals across 73 countries and the increasing incidence of microcephaly in newborns led the WHO to declare Zika as a public health emergency of international concern [65,66]. That a further 35 countries recorded imported cases underlines the facility with which viruses can be spread globally at a rapid pace. Outbreaks of chikungunya now occur frequently across the globe in all continents with the numbers of cases reaching hundreds of thousands [67]. In addition to this major tropical mosquito vector species, novel invasive mosquito spp. competent for arbovirus transmission have spread geographically. This is the case for *Aedes albopictus*, which spread from South East Asia into Europe and the United States in the 1990s/2000s [68] and is competent for transmitting DENV, Zika virus, and chikungunya virus [69]. Whilst its persistence in temperate climates reflects its capacity to overwinter at the egg stage, warmer spring, summer and autumnal months will enable an increased vectorial capacity. Similarly, two other *Aedes* spp., *Aedes japonicus* and *Aedes koreicus*, both of which, like *Ae. albopictus*, have desiccation-resistant *eggs*, which can serve as an overwintering mechanism, have invaded Europe, and *Ae. japonicus* has invaded the USA [70,71]. Both of these species have been incriminated in Japanese Encephalitis virus transmission, another member of the flaviviridae [72]. *Ae. albopictus* and *Ae. japonicus* have also been incriminated in transmission of La Crosse encephalitis virus (a bunyavirus) in the USA [73]. In addition, several other *Aedes* spp. incriminated as vectors for arboviruses and parasitic nematodes have shown recent evidence of territorial expansion. *Aedes notoscriptus*, a major vector of the Ross River virus and *Dinofilaria* in Australasia, has recently been found in California [74,75]. *Aedes scapularis*, vectors of arbovirus and filaria, has spread north from the neotropics to California and Florida [76]. Finally *Aedes vittatus*, a widely distributed vector in the Old World and incriminated in the transmission of Zika virus, has now been identified in Cuba and the Dominican Republic [77,78,79]. All of these six invasive species are container-breeding and can, to varying extents, exploit urban habitats [80,81].

As well as the increase in the spread of invasive mosquito vectors, novel arboviruses are also spreading into naïve areas that have pre-existing competent mosquito vectors. The most well documented case is the arrival and expansion of West Nile virus, another flavivirus, in the USA. First identified in an epidemic in New York in 1999, the virus that predominantly infects avian and equine spp., has now spread throughout the continent and is now considered the virus with the greatest global distribution [82,83]. The main vectors of this virus are *Culex pipiens* complex mosquitoes and *Culex tarsalis*. *Cx. pipiens* complex mosquitoes are well adapted to ovipositing in stagnant water, including places with poor drainage, urban catch roadside ditches and manmade containers around houses [84]. This high proximity of productive mosquito breeding sites to human urban populations can generate very high numbers of cases at a very local scale in a manner very similar to St. Louis Encephalitis case clustering (a flavivirus transmitted by the same mosquito vector spp. amongst others) [85,86].

Urban malaria has been estimated to account for 6%–28% of the global annual disease incidence, predominantly reflecting the large increase in urbanization in Africa and subsequent increase in peri-urban transmission [87]. In addition, there is some evidence that the major African mosquito vector of malaria parasite spp., *Anopheles gambiae*, has adapted to ovipositing in artificial containers [88]. Formerly, urban malaria was an epidemiological facies unique to settings where *An. stephensi* was present. This mosquito species, like *Ae. aegypti*, has adapted to an urban environment, utilizing water storage containers as larval habitats [89,90]. Formerly restricted to South East Asia, India and the Arabian peninsula prior to 2011, it has now been reported from Djibouti (2012), Ethiopia (2016), Sri Lanka (2017), and most recently from the Republic of the Sudan (2019) [91,92,93]. This species is well capable of thriving in an urban environment and is thus threatening to bring an additional malaria burden to that driven by the more rural mosquito vector spp. in a world of ever-increasing urbanization.

In light of the increasing number of urban mosquito spp. and pathogens spreading geographically and the relatively paucity of our knowledge of key epidemiological drivers, curbing the trajectories of the major urban vector-borne health problems will be a challenge. Beyond integrated vector management, what mitigation strategies can be taken or planned for within the general context of making cities healthier. How to articulate mosquito-borne disease control with all the other health problems that cities face?

## 4. Mitigation Strategies for UHIs

In recognition of the growing and predicted problem of UHIs, a considerable effort has been made to develop mitigation strategies. The most extensive approach has been undertaken by Singapore, which has developed over 80 different strategies [94]. These can be grouped into seven categories: vegetation, urban geometry, water bodies, materials and surfaces, shading, transport, and energy. Strategies based around vegetation have been highlighted as particularly effective, as well as being cost effective and relatively easily and rapidly implemented as compared with the other categories. Hence vegetation has been used extensively as a UHI mitigation strategy worldwide [95,96,97] in innumerable ways at micro (e.g., vertical gardens, planting trees along transport axes creating green corridors) and mesoscales (e.g., increasing urban parks and encouraging urban agriculture). Planting vegetation reduces the impact of incoming solar energy by shading and because the relatively high albedo of vegetation also reduces the accumulation of heat. The shading effect can not only reduce the sensible heat flux but also the latent heat flux. Green corridors can not only shade via trees, but also the use of grass reduces turbulent vertical air movements produced by hot surfaces, thereby improving convection efficiency. One problem is, however, that in tropical countries with high humidity, vegetation can increase thermal discomfort. In such wet climates, the extent of green coverage necessary to reduce the temperature differential (of the UHIs compared to rural areas) would lead to excessive thermal discomfort. Hence, it has been suggested that under such circumstances, increased shading rather than simply increased vegetal mass would be more effective [98,99]. By contrast, in drier climates that are water-limited, there is low evapotranspiration and low albedo, and thus increased greening is predicted to be more effective [98,99]. This underlines the point that strategies will not be expected to work the same everywhere.

Another problem with excess vegetation is that it impacts upon another health issue: mosquito-borne disease. Inner city parks and green zones can considerably reduce local surrounding temperatures [100], as well as providing habitats for the less urbanized mosquito species, such as those invasive species listed previously. A prime example of this outside of the tropics is the dengue outbreak in Tokyo with its epicenter in Yoyogi Park and vectored by *Ae. albopictus* mosquitoes [101]. Inner city parks have also been found to be associated with increased dengue risk for people living in dwellings close to such green spaces even where the urban vector *Ae. aegypti* predominates [102]. Such green spaces may therefore provide a safe haven for adult mosquitoes of all species in the face of extreme urban temperatures. Additionally, in light of the temperature-dependency of the mosquito vectorial capacity for arboviruses and malaria parasites, cooling green zones may lead to unexpected impacts on transmission rates [103]. Optimal temperatures for such pathogen development and transmission tend to range around 25 °C–35 °C depending on humidity and diurnal temperature ranges [103]. Very warm temperatures in the tropics can thus generate sub-optimal conditions and thus green cooling zones may inadvertently improve conditions for the pathogen. Therefore, transmission could be enhanced by green zones, both from a perspective of increased habitat for the less urbanized invasive species and for cooling for the vectorial capacity. Such impacts will change according to the climate zone within which cities are located and the ecology of the local community of mosquito spp. Hence, differential effects of UHI mitigation on pathogen transmission can be anticipated in different places.

## 5. Tackling Urban Mosquito-Borne Disease in the Context of UHI and Other Sectoral Climate Change Adaptations

With the prospect of UHI mitigation strategies being increasingly developed globally to combat, in particular, the non-infectious disease burden, how to evolve current mosquito control strategies? Mosquito control in urban settings is very challenging, not least because of the complexity of the environment and the multitude of man-made containers that can provide oviposition sites for mosquitoes to lay eggs in [89]. Current integrated vector management approaches focus on source reduction by improved environmental hygiene (eliminating solid waste that offer potential oviposition sites), using larvicides in stored water objects (vases, urns, storage jars, overhead tanks etc.) to eliminate the immature stages of the mosquitoes and adulticides to kill adult stages of the mosquito. This latter is generally employed in and around houses of clinical cases identified through the public health system. Source reduction and community-based environmental management have shown some success [104,105], but requires massive efforts and considerable community engagement. Personal protective methods such as window netting and mosquito nets, coils, and sprays are actively encouraged, but whose efficacy against daytime biting mosquitoes is unclear [106]. However, in response to the growing global threat imposed by vector-borne diseases and in recognition that current methods are inadequate, not least for urban diseases, the World Health Organization launched its new initiative, the 2017–2030 Global Vector Control Response to galvanize efforts [107]. Whilst much of the focus is placed on the vectors of current major importance (such as *Ae. aegypti*), within the special context of the urban setting and with increasing recognition of the importance of the “new” invasive spp. and the concomitant promotion of green spaces, additional intervention methods need to be envisaged [108].

One of the major challenges for targeting the immature mosquito stages is the inability to find and treat the abundance of potential breeding sites. One new method developed for *Ae. aegypti* is to use the adult female mosquitoes themselves to disseminate the insecticide–auto-dissemination–with the assumption that she will find oviposition sites better than humans [109,110,111,112]. Whilst this has been to shown to work for both *Ae. aegypti* and *Ae. albopictus*, at least in reducing mosquito numbers, the urban geometry has a significant impact on efficacy, for the simple reason that mosquitoes do not fly over high walls and generally will not fly very far [113,114,115]. This hampers the dissemination of the insecticide and requires deployment of auto-dissemination devices at unacceptably high densities. Whilst other new control methods are showing promise [116], mitigating against urban vector-borne diseases could use the leverage of other sectors impacted upon by climate change–notably water, environmental/ecosystem hygiene and urban structural improvements.

Managing water resources is predicted to become a serious problem, especially in the urban setting and inadequate permanent access to water leads to water storage practices that enables proliferation of mosquito populations and increased disease [32]. Lack of piped water has been shown to be a risk factor for dengue and is likely also to be the case for urban malaria [32]. In this case, strategies to improve water supplies could lead to sectorial co-benefits. It is also a clear concern for drinking water quality generally. Inter-sectoral co-benefits would thus arise from focusing on access to water. Improved environmental hygiene is crucial for reducing pollutants and maintaining a healthy urban environment. Solid waste matter provides abundant sites for mosquito breeding [117]. Thus again, strategies focused on environmental hygiene would lead to inter-sectoral co-benefits. Urban housing quality and geometry impact upon the extent of exposure of man to mosquitoes and are clear determinants of heat impacts. Urban planning does to some extent now include this heat aspect of health, but considerations of creating a mosquito-proof environment remain neglected. That housing quality, with specific respect to exposure to mosquitoes, is an important factor for protection against arboviral infection has been demonstrated by several studies [118,119]. Good housing quality and air-conditioning were associated with reduced exposure to DENV in the US on the Texas-Mexican border despite higher mosquito densities on the US side [118]. Insecticide treated screens have also been shown to reduce infection rates with Zika virus and offer a potentially simpler housing improvement than heat-generating, energy-consuming air-conditioners [119]. Even more imaginative ideas could be proposed and tested. For example, combining mosquito control methods such as auto-dissemination and use of toxic baited sugar traps [120] could benefit from the creation of green oases that lure mosquitoes to these safe havens and either locally kill the adult mosquitoes or enable the adults to disseminate the insecticide more freely along green “ventilation” corridors [121]. Whilst there will not be one single suite of integrated vector control methods that are applicable to all mosquito spp. everywhere, multi-sectoral collaborations in designing healthier cities must be actively encouraged and pursued. Improved access to permanent running water, reduced urban heat stress through improved urban geometry amongst other approaches, and a reduced burden of mosquito-borne disease need to be considered through multi-sectorial collaboration for co-creation of resilient strategies [122,123].

## 6. Conclusions

Despite the fundamental importance for public health to know the magnitude and patterns of climate impacts and their significance for prioritization and allocation of resources to protect populations, the health research community largely work in isolation of the other sectors. This is equally true within the health sector for the non-infectious vs. infectious health issues. However, to ensure sustainable development, it is imperative that all aspects of urban public health are considered in climate change policies and climate services across sectors. Multi-sectoral collaboration has improved over recent years through the Inter-Sectorial Impact Model Inter-comparison Project, but health has been, unlike other sectors, only vaguely and sporadically present in this effort [124,125,126]. Such collaborative multi-sectoral work now needs to be carried out at the specific level of cities to align with the sustainable development goals 3, 6, and 11: Ensuring good health and well-being, universal access to clean water and sanitation and making cities and human settlements inclusive, safe, resilient, and sustainable through improvements to housing and basic services [123]. There are a significant number of mitigation strategies that would allow cities to confront the health challenges of climate change. Multi-sectoral collaboration would optimize the meagre resources with which cities have to manage the many foreseeable problems that will arise.

## Data Availability

Not applicable.

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
