# Peer review of "Challenges to Mitigating the Urban Health Burden of Mosquito-Borne Diseases in the Face of Climate Change"

_ijerph, 2021, doi:10.3390/ijerph18095035_

Round 1
Reviewer 1 Report
The manuscript has improved. As I mentioned in my previous review, information regarding global warming and urban heat islands is really interesting. However, I have some concerns in some parts of the ms.
Line 94-100 could be deleted. It is really interesting information, however, is distractive. The main topic is vector borne diseases.
Section 4 and 5 have some weaknesses:
New ideas are always welcome, however, this need an integrative knowledge about vectors and vector-borne diseases. Authors should have in mind two related points: (1) current settlements with a long history of vector borne disease and new settlements at risk of vector borne disease. The strategies for mosquito management might different, control vs. prevention (and/or mixed strategy). (2) Mosquito biology: (complete) endemic urban mosquitoes (e.g. Aedes aegypti, Culex spp) vs. endemic semi-urban/sylvatic mosquitoes (some Aedes, Anopheles, Culex spp). Will Aedes aegypti (the most urban mosquito worldwide) really be affected by implementing the mitigation strategies mentioned on section 4 in current and new settlements? The urban biology of the species has taught us that the mosquito does not need to fly great distances to have a food source (humans) and breeding sites (containers) the life cycles usually is in the same areas (house); Aedes aegypti does not need green areas to complete its life cycle. So creating green areas or corridors will not really have an impact on Aedes aegypti. Maybe these strategies could be useful against semi-urban/sylvatic mosquitoes.
So, section 4 did not really connect with section 5. Some control measures/strategies mentioned on section 5 are the current strategies used or have been considered for reducing mosquito population densities without considering global warming or UHIs.
Authors should have in mind the previous points and rewrite section 4 and 5.
Reviewer 2 Report
This manuscript has been thoroughly revised, and I am satisfied with the authors’ refocusing – well done. This new version reads very well. I have a few very minor comments and suggestions:
The new title is a little wordy and awkward – I suggest rewording this, perhaps “Challenges to mitigating the urban health burden of mosquito-borne diseases in the face of climate change”
L64: A relevant recent publication that could be mentioned here, if the authors find it useful: Dialesandro et al. 2021, Dimensions of Thermal Inequity: Neighborhood Social Demographics and Urban Heat in the Southwestern U.S., IJERPH 18:941.
L102: the second “will” on this line should be “with”, correct?
L127: suggest inserting “invasive mosquito vectors and” after “importation of”
L141: “worse” should be “worst”, unless it should be “best”, as in “even under the best-case scenario”
L179: Consider mentioning here introductions of the vector mosquitoes Aedes notosciptus and Aedes scapularis into the US (CA and FL, respectively), and Aedes vittatus into the Caribbean.
L210: Should this sentence end in a question mark?
L211: Awkward wording, suggest rewording.
Round 2
Reviewer 1 Report
This will be an usefull review.
This manuscript is a resubmission of an earlier submission. The following is a list of the peer review reports and author responses from that submission.
Round 1
Reviewer 1 Report
The manuscript “The Challenge of Controlling Urban Mosquito-Borne Diseases in a Changing Urban Climate” by Ligsay et al. describes the issue of the relationship between climate change, urbanization and mosquito-vectored disease. It is well written, but it is rather unfocused, and does not thoroughly review the issue defined in the title, and in this reviewers opinion, in its current state, it does not constitute a review of this topic. Given the title, little of the manuscript addresses the challenge of controlling mosquito-vectored disease in urban areas, with much of the manuscript being focused on describing climate change history and climate change impacts. While this information is mostly relevant, this is the bulk of the manuscript, with comparatively little focus on mosquito-borne disease and its control. Concomitantly, some of this information is irrelevant to the topic stated in the title. For example, Box 1 could be entirely excluded as this information is not utilized elsewhere in the manuscript, and is not presented in the context of mosquito-vectored pathogens.
I suggest that the authors improve the focus of this manuscript by elaborating on section 4 and 7, removing extraneous background information in sections 1, 2, 3, 5, and 6 (for example, Box 1, Section 3. Climate change impacts on health as this section is focused on health impacts irrelevant to mosquito-vectored pathogens, and other sections that are peripheral to the topic defined by the title), and improving the focus of the manuscript to the topic defined in the title. Alternatively, the authors could adjust the title to better fit the manuscript. As it is though, the focus of this manuscript is not narrow enough on control of urban mosquito-borne disease to constitute a review of this topic.
The authors should strongly consider elaborating on sections 4 and 7, and making these sections more thorough as these sections are most relevant to the topic defined in the title. For example, a major concern related to the impact of climate change on mosquito-vectored pathogens that is not covered in this review is that climate change can affect the geographic distribution of mosquito vectors, and mosquito species are increasingly being moved to new locations and establishing populations. This relates to control in urban areas as new mosquito vectors such as Aedes aegypti, Aedes albopictus, Aedes japonicus and Aedes scapularis are being moved to new locations where they were previously absent, and in many cases, becoming common within cities, thereby, contributing to the challenges associated with vector-borne disease in urban areas.
In general, there is a high level of detail on topics peripheral to the challenge of controlling urban mosquito-vectored pathogens with regard to climate change, and a rather limited level of detail on topics more narrowly associated with these challenges.
Minor comments:
L17: delete “such”
L31: Agriculture should not be capitalized
L50: “as” should be “at”
Introduction: This section does not include any mention of mosquito-vectored pathogens.
L83: “has” should be “have”
L100: “will” should be “with”
L113/114: Delete either “notably” or “including”
L114 and throughout manuscript: Initial letter on chikungunya should not be capitalized.
L121-122: Mosquito-borne pathogens are not limited to the tropics and subtropics. E.g., West Nile virus, eastern equine encephalitis virus, St. Louis encephalitis virus, etc.
L129: Reference 41 is not appropriate here. Hawaii is within the tropics. There are many references on dengue outbreaks in Florida and elsewhere in the US. May be relevant to state also, that historically, there have been widespread outbreaks of dengue in the southeastern US, and that yellow fever and malaria were historically prevalent in the temperate eastern US.
L131: Suggest discussing the spread of Aedes albopictus.
L132: “Yellow Fever” should be “yellow fever”
L143: italicize Anopheles
L146: suggest changing “breeding in water storage containers” to “utilizing water storage containers as larval habitats”
Reviewer 2 Report
Overall, this is a well written article with a lot of interesting information on climate change into urban environment. However, the article is more about the challenges posed by the climate change and mosquito-borne diseases are taken as just one example of the several health issues posed by climate change. I would suggest changing the title to better reflect the content of the article such as something like: "The challenges posed by Climate change to public health into urban environment through some example from mosquito-borne diseases".
Minor comments are included in attached file.

Reviewer 3 Report
In their paper, Ligsay et al, tried to explain the relationship between the control of urban mosquito-borne diseases (VBDs) and climate change. Unfortunately, the purpose of and problems to solve in the work are not clearly stated. As a result, real, interesting and important conclusions are lacking. The presentations of the ideas are not satisfactory, and the title does not really reflect the contents of the paper. Abstract did not describe the essential information in the work (mainly because is missing). There is not introductory section to adequately explain the framework and problems of the research.
Valuable and interesting information regarding climate change and impact on different aspects is given. However, the ms as it is written, can be related to any disease, vector-borne or not vector-borne diseases, transmissible and not transmissible diseases. Through all the ms there is not real connection between vector control problematic and global warming, climate change and projection scenarios, or heat island. Box 1 is also not related to VBDs. Lines 112-175 are the only interconnected information of climate change and vector-borne diseases.
There is important information missing:
Reiter P. (2001). Climate change and mosquito-borne disease. Environmental health perspectives, 109 Suppl 1(Suppl 1), 141–161. https://doi.org/10.1289/ehp.01109s1141
Epstein, P. R., Diaz, H. F., Elias, S., Grabherr, G., Graham, N. E., Martens, W. J. M., MosIey-Thompson, E., & Susskind, J. (1998). Biological and Physical Signs of Climate Change: Focus on Mosquito-borne Diseases, Bulletin of the American Meteorological Society, 79(3), 409-418.
Franklinos LHV, Jones KE, Redding DW, Abubakar I. The effect of global change on mosquito-borne disease. Lancet Infect Dis. 2019 Sep;19(9):e302-e312. doi: 10.1016/S1473-3099(19)30161-6.
Rocklöv, J., Dubrow, R. Climate change: an enduring challenge for vector-borne disease prevention and control. Nat Immunol 21, 479–483 (2020)
Araujo, Ricardo Vieira et al. São Paulo urban heat islands have a higher incidence of dengue than other urban areas. Braz J Infect Dis [online]. 2015, vol.19, n.2 [cited 2021-03-11], pp.146-155.
de Azevedo, T. S., Bourke, B. P., Piovezan, R., & Sallum, M. A. M. (2018). The influence of urban heat islands and socioeconomic factors on the spatial distribution of Aedes aegypti larval habitats. Geospatial Health, 13(1).